# Elimination of QCD Renormalization Scale and Scheme Ambiguities

Sheng-Quan Wang [1], Stanley J. Brodsky [2,*], Xing-Gang Wu [3,*], Jian-Ming Shen [4] and Leonardo Di Giustino [5,6]

[1] Department of Physics, Guizhou Minzu University, Guiyang 550025, China
[2] SLAC National Accelerator Laboratory, Stanford University, Stanford, CA 94039, USA
[3] Department of Physics, Chongqing University, Chongqing 401331, China
[4] Hunan Provincial Key Laboratory of High-Energy Scale Physics and Applications, School of Physics and Electronics, Hunan University, Changsha 410082, China
[5] Department of Science and High Technology, University of Insubria, Via Valleggio 11, 22100 Como, Italy
[6] INFN, Sezione di Milano-Bicocca, 20126 Milano, Italy
[*] Correspondence: sjbth@slac.stanford.edu (S.J.B.); wuxg@cqu.edu.cn (X.-G.W.)

**Abstract:** The setting of the renormalization scale ($\mu_r$) in the perturbative QCD (pQCD) is one of the crucial problems for achieving precise fixed-order pQCD predictions. The conventional prescription is to take its value as the typical momentum transfer $Q$ in a given process, and theoretical uncertainties are then evaluated by varying it over an arbitrary range. The conventional scale-setting procedure introduces arbitrary scheme-and-scale ambiguities in fixed-order pQCD predictions. The principle of maximum conformality (PMC) provides a systematic way to eliminate the renormalization scheme-and-scale ambiguities. The PMC method has rigorous theoretical foundations; it satisfies the renormalization group invariance (RGI) and all of the self-consistency conditions derived from the renormalization group. The PMC has now been successfully applied to many physical processes. In this paper, we summarize recent PMC applications, including event shape observables and heavy quark pair production near the threshold region in $e^+e^-$ annihilation and top-quark decay at hadronic colliders. In addition, estimating the contributions related to the uncalculated higher-order terms is also summarized. These results show that the major theoretical uncertainties caused by different choices of $\mu_r$ are eliminated, and the improved pQCD predictions are thus obtained, demonstrating the generality and applicability of the PMC.

**Keywords:** perturbative QCD; QCD renormalization scale; principle of maximum conformality; event shapes

## 1. Introduction

The asymptotic freedom property of quantum chromodynamics (QCD) was proposed by Politzer, Gross, and Wilczek 50 years ago [1,2]. Due to the asymptotic freedom property, the strong interaction whose magnitude can be characterized by the QCD strong coupling $\alpha_s$ becomes small at very short distances, allowing perturbative calculations for observables involving large momentum transfer. The strong coupling $\alpha_s$ is scale-dependent, and its behavior is controlled by renormalization group equation (RGE),

$$\beta(\alpha_s) = \frac{d}{d\ln\mu_r^2}\left(\frac{\alpha_s(\mu_r)}{4\pi}\right) = -\sum_{i=0}^{\infty}\beta_i\left(\frac{\alpha_s(\mu_r)}{4\pi}\right)^{i+2}. \tag{1}$$

The $\beta$ functions $\beta_0$, $\beta_1$, $\cdots$ are one-loop, two-loop, $\cdots$ corrections, respectively.

In the framework of perturbative QCD (pQCD), the prediction for an observable $\rho$ at the $n_{\text{th}}$-order level can be expressed as a perturbative series over the QCD coupling $\alpha_s$, i.e.,

$$\rho = \sum_{i=0}^{n} C_i \, \alpha_s(\mu_r)^{p+i}, \tag{2}$$

where $p$ is the power of the $\alpha_s(\mu_r)$ for the tree-level terms. The scale $\mu_r$ represents the initial choice of renormalization scale. The coefficients $C_1$, $C_2$, $\cdots$ are one-loop, two-loop, $\cdots$ corrections, respectively. The pQCD predictions, calculated up to all orders with $n \rightarrow \infty$, are independent of the choice of the renormalization scheme and renormalization scale because of the renormalization group invariance (RGI). At any finite order, the renormalization scheme and scale dependence of the QCD coupling constant $\alpha_s(\mu_r)$ and of the QCD perturbative coefficients $C_i$ only partially cancel. For example, it has been conventional to guess the renormalization scale $\mu_r$, choosing among the typical scales of a process, e.g., the typical momentum transfer $Q$, in order to minimize large logarithmic corrections and achieve relativistically more convergent series. This conventional procedure breaks the RGI and introduces renormalization scheme-and-scale ambiguities in pQCD predictions. The conventional scale-setting method also has the negative consequence that the resulting pQCD series suffers from a divergent renormalon ($\alpha_s^n \beta_0^n n!$) series [3] characteristic of a non-conformal series at order $n$. Furthermore, theoretical uncertainties estimated by simply varying the renormalization scale $\mu_r$ over an arbitrary range such as $\mu_r \in [Q/2, 2Q]$ are clearly unreliable, since they are only sensitive to the $\beta$-dependent non-conformal terms, not the entire pQCD series. One cannot judge whether the slow convergence is an intrinsic property of pQCD series or is due to the improper choice of renormalization scale $\mu_r$.

The conventional scale-setting procedure is also inconsistent with the well-known Gell-Mann-Low (GM-L) method used in QED [4]. In practice, the GM-L method shows that, by fixing the scale to the correct momentum flow, one can reabsorb all the vacuum polarization diagrams into the running coupling. Thus, the renormalization scale-setting procedure in QED is void of any ambiguity. A self-consistent scale-setting procedure should be adaptable to both QCD and QED. In the limit of $N_C \rightarrow 0$ [5], predictions for non-Abelian QCD theory must agree analytically with predictions for Abelian QED, and this also includes the renormalization scale-setting procedure. Thus, the elimination of the ambiguities in order to achieve precise pQCD predictions is crucial for testing the standard model (SM) and for searching for new physics beyond the SM.

The well-known Brodsky–Lepage–Mackenzie (BLM) method has been suggested in Ref. [6] and has been improved to all orders as the principle of maximum conformality (PMC) [7–11] method. The PMC is the underlying principle for the BLM method and provides a systematic all-orders way to eliminate the renormalization scheme-and-scale ambiguities. This method extends the BLM procedure unambiguously to all orders, to all processes, and to all gauge theories. The PMC method meets all the rigorous theoretical requirements, satisfying both the RGI [12–14] and the self-consistency conditions derived from the renormalization group [15]. The PMC method reduces to the GM-L method in the Abelian limit. A remarkable achievement of the PMC is that the resulting scale-fixed predictions for physical observables are independent of the choice of renormalization scheme—a key requirement of RGI.

In 2017, the PMC single-scale method (PMCs) [16] was suggested, which is is equivalent to the multi-scale method [7–11] in the sense of perturbative theory. The PMCs method effectively replaces the individual PMC scales at each order derived by using the PMC multi-scale method in the sense of a mean value theorem. In 2020, we used an additional property of renormalizable SU(N)/U(1) gauge theories [17], "Intrinsic Conformality (iCF)", which underlies the scale invariance of physical observables. It shows that the scale-invariant perturbative series shows the intrinsic perturbative nature of a pQCD observable. In 2022, following the idea of iCF, we suggested a novel single-scale-setting method under the PMC with the purpose of removing the conventional renormalization scheme-and-scale ambiguities [18]. In Ref. [18], it has been demonstrated that the two PMCs methods are equivalent to each other in accuracy. This equivalence indicates that, by

using the RGE for fixing the value of the effective coupling, it is equivalent to requiring that each loop term must satisfy the scale invariance simultaneously, and vice versa. Thus, using the RGE provides a rigorous way to resolve conventional scale-setting ambiguities.

## 2. A Mini-Review of the PMC Scale-Setting Method

The scale evolution of $\alpha_s$ is described by the RGE as shown by Equation (1), which can be used recursively to establish the perturbative pattern of $\{\beta_i\}$-terms at each order. The pQCD prediction for a physical observable $\rho$ can be reorganized into the specific "degeneracy" pattern [19] as follows:

$$
\begin{aligned}
\rho(Q) \;=\; & r_{1,0}\,\alpha(\mu_r)^p + (r_{2,0} + p\beta_0 r_{2,1})\,\alpha(\mu_r)^{p+1} + \left( r_{3,0} + p\beta_1 r_{2,1} + (p+1)\beta_0 r_{3,1} + \frac{p(p+1)}{2}\beta_0^2 r_{3,2} \right)\alpha(\mu_r)^{p+2} \\
& + \left( r_{4,0} + p\beta_2 r_{2,1} + (p+1)\beta_1 r_{3,1} + \frac{p(3+2p)}{2}\beta_1\beta_0 r_{3,2} + (p+2)\beta_0 r_{4,1} + \frac{(p+1)(p+2)}{2}\beta_0^2 r_{4,2} \right. \\
& \left. + \frac{p(p+1)(p+2)}{3!}\beta_0^3 r_{4,3} \right)\alpha(\mu_r)^{p+3} + \cdots ,
\end{aligned}
\tag{3}
$$

where $\alpha = \alpha_s/4\pi$ and $Q$ represents the kinematic scale at which the observable is measured. The coefficients $r_{i,0(i=1,2,\cdots)}$ are conformal parts and $r_{i,j(i>j\geq1)}$ are non-conformal ones. All the non-conformal coefficients $r_{i,j(i>j\geq1)}$ are, in principle, functions of the scales $\mu_r$ and $Q$.

Following the PMC multi-scale procedures [7–11], all the non-conformal $\{\beta_i\}$-terms in Equation (4) are systematically eliminated to fix the correct magnitudes of QCD running couplings at each order (their arguments are referred to as PMC scales); the resulting perturbative series then matches the corresponding conformal theory with $\beta = 0$, leading to scheme-independent predictions. The divergent renormalon contributions are eliminated, and the convergence of the perturbative series is in general greatly improved. This is the same principle used in QED where all $\{\beta_i\}$-terms derived from the vacuum polarization corrections of the photon propagator are absorbed into the QED coupling. As in QED, the resulting PMC scales are physical in the sense that they reflect the virtuality of the gluon propagators at a given order, and that they set the active flavors $n_f$. More explicitly, after applying the PMC multi-scale method, the pQCD series for the physical observable $\rho$ becomes

$$
\begin{aligned}
\rho(Q) \;=\; & r_{1,0}\,\alpha(Q_1)^p + r_{2,0}\,\alpha(Q_2)^{p+1} + r_{3,0}\,\alpha(Q_3)^{p+2} \\
& + r_{4,0}\,\alpha(Q_4)^{p+3} + \cdots ,
\end{aligned}
\tag{4}
$$

where $Q_{i=1,2,3,4}$ are the PMC scales. Due to uncalculated higher-order contributions, there are two kinds of residual scale dependences [20]. The first kind of residual scale dependence is from the PMC scale itself because the PMC scale is a perturbative expansion series in $\alpha_s$. The second kind of residual scale dependence is from the last terms of the pQCD approximant because its magnitude cannot be determined. These residual scale dependencies are distinct from the conventional renormalization scale ambiguities and are suppressed due to the perturbative nature of the PMC scale.

In order to suppress the residual scale dependence, which also makes the PMC scale-setting procedures simpler and more easily automatized, the PMCs method has been suggested in Ref. [16]. The PMCs method provides a self-consistent way to achieve precise $\alpha_s$ running behavior in both the perturbative and nonperturbative domains [21,22]. After applying the PMCs method, the pQCD prediction for the physical observable $\rho$ can be written as

$$
\begin{aligned}
\rho(Q) \;=\; & r_{1,0}\,\alpha(Q_\star)^p + r_{2,0}\,\alpha(Q_\star)^{p+1} + r_{3,0}\,\alpha(Q_\star)^{p+2} \\
& + r_{4,0}\,\alpha(Q_\star)^{p+3} + \cdots .
\end{aligned}
\tag{5}
$$

The single PMC scale $Q_\star$ is determined by requiring all the non-conformal $\{\beta_i\}$-terms to vanish simultaneously and can be regarded as the overall effective momentum flow of the

process. The PMCs method exactly removes the second kind of residual scale dependence, and the first kind of residual scale dependence is highly suppressed. The PMCs method eliminates the renormalization scheme-and-scale ambiguities and satisfies the standard RGI [14].

Up to now, the PMC approach has been successfully applied to many physical processes (see, e.g., [13,14,23] for reviews), including the Higgs boson production at the LHC, the Higgs boson decays to $\gamma\gamma$, $gg$, and $b\bar{b}$ processes, the top-quark pair production at the LHC and Tevatron and its decay process [24], the semihard processes based on the BFKL approach [20,25–27], the electron–positron annihilation to hadrons [10,11,13], the hadronic $Z^0$ boson decays, the event shapes in electron–positron annihilation [17,28], the electroweak parameter $\rho$ [29,30], the $\Upsilon(1S)$ leptonic decay [31,32], and the charmonium production [33–35]. In addition, the PMC provides a possible solution to the $B \to \pi\pi$ puzzle [36] and the $\gamma\gamma^* \to \eta_c$ puzzle [37]. In the following, we present some recent PMC applications and a way of estimating unknown contributions from uncalculated higher-order terms by using the PMC pQCD series.

## 3. Applications

### 3.1. New Analyses of Event Shape Observables in $e^+e^-$ Annihilation

Event shapes represent an ideal platform for high-precision tests of QCD (see e.g., [38] for a summary from Particle Data Group). The experiments at LEP and at SLAC have measured event shape distributions with very high precision, especially those at the $Z^0$ peak [39–43]. On the theoretical side, the pQCD corrections to event shape observables have been calculated up to the next-to-next-to-leading order (NNLO) [44–50]. Currently, one finds that the main obstacle for achieving highly precise measurements of $\alpha_s$ from event shape variables is given by theoretical uncertainties, especially those related to the renormalization scale ambiguities.

Comprehensive PMC analysis for event shapes in $e^+e^-$ annihilation and a novel method for the precise determination of the QCD running coupling $\alpha_s(Q^2)$ are shown in Refs. [28,51]. Interested readers may turn to these studies for more details. In this paper, we only present the main PMC results for two fundamental event shapes: the thrust ($T$) [52,53] and the $C$-parameter ($C$) [54,55].

In the case of conventional scale setting, one simply sets the renormalization scale $\mu_r$ to the center-of-mass energy $\mu_r = \sqrt{s}$. We present the thrust and $C$-parameter differential distributions using the conventional scale-setting method at $\sqrt{s} = 91.2$ GeV in Figure 1. Results show that even up to NNLO QCD corrections, the conventional results are plagued by large-scale uncertainty and substantially deviate from the precise experimental data.

Moreover, the method does not improve the precision at higher orders, since the results are totally arbitrary. In fact, varying the $\mu_r \in [\sqrt{s}/2, 2\sqrt{s}]$, the NLO calculation does not overlap with the LO prediction, and the NNLO calculation does not overlap with the NLO prediction. This indicates that the evaluation of uncalculated higher-order (UHO) terms for event shape observables by varying $\mu_r \in [\sqrt{s}/2, 2\sqrt{s}]$ is not quantitatively reliable. Worse, since the renormalization scale is simply set to $\mu_r = \sqrt{s}$, only one value of $\alpha_s$ at the scale $\sqrt{s}$ can be extracted, with an arbitrary large error given by the choice of the renormalization scale $\mu_r$.

The PMC scales are determined by absorbing all the $\beta$ terms of the pQCD series. In Figure 2, we show the PMC scales for thrust and $C$-parameter at the scale $\sqrt{s} = 91.2$ GeV. The resulting PMC scales are not a single value, but they monotonically increase with the value of $T$ and $C$, reflecting the increasing virtuality of the QCD dynamics. The number of active flavors $n_f$ changes with the value of $T$ and $C$ according to the PMC scales. It is noted that the quarks and gluons have soft virtuality near the two-jet region (left boundary). As the argument of the $\alpha_s$ approaches the two-jet scale-region, the PMC scales are very soft. Thus, the dynamics of the PMC scale reflect the correct physical behavior when approaching the two-jet region. In addition, the PMC scales are small in the wide kinematic regions compared to the conventional choice of $\mu_r = \sqrt{s}$.

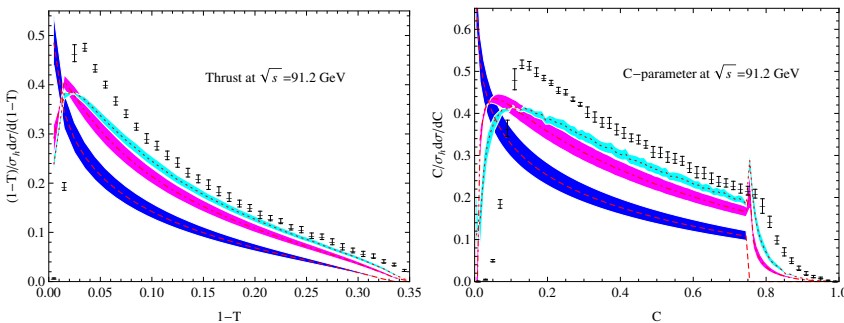

**Figure 1.** The thrust (*T*) and *C*-parameter (*C*) distributions using the conventional scale-setting method at $\sqrt{s} = 91.2$ GeV, where the dashed, dot-dashed, and dotted lines are the conventional results at LO, NLO, and NNLO [45,48], respectively. The bands are obtained by varying the scale $\mu_r \in [\sqrt{s}/2, 2\sqrt{s}]$. The experimental data are taken from the ALEPH Collaboration [39].

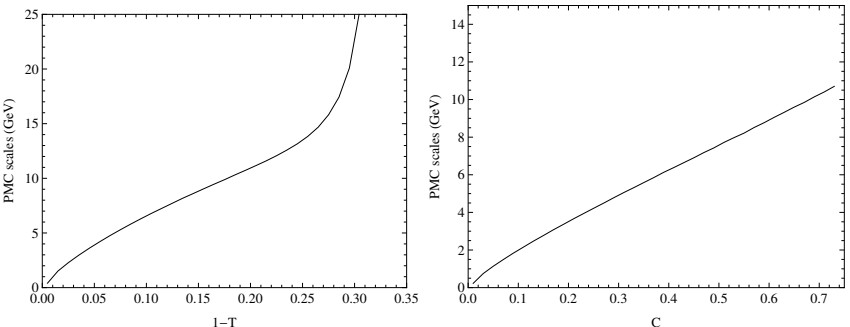

**Figure 2.** The PMC scales for the event shape observables thrust (*T*) and *C*-parameter (*C*) at $\sqrt{s} = 91.2$ GeV.

It is noted that the behavior of the PMC conformal coefficients is significantly different from the pQCD terms given by the conventional scale-setting method. Since the conformal coefficients are renormalization scale-independent, the resulting PMC predictions eliminate the renormalization scale uncertainty. By setting all input parameters to their central values, we present the thrust and *C*-parameter distributions using the PMC scale-setting method for $\sqrt{s} = 91.2$ GeV in Figure 3. This figure shows that the PMC predictions improve for a wide range of values in the kinematic regions with respect to the conventional scale-setting predictions and are in excellent agreement with the experimental data, especially in the intermediate kinematic regions. Since there are large logarithms that spoil the perturbative regime of the QCD near the two-jet region and there are missing higher-order contributions that are important for the multi-jet region, the PMC predictions in these regions show some deviations from experiments.

The resummation of large logarithms is thus required for the PMC predictions, especially near the two-jet region. In fact, the resummation of large logarithms has been extensively studied in the literature.

For the extraction of $\alpha_s$, since the renormalization scale is simply set as $\mu_r = \sqrt{s}$ when using conventional scale setting, only one value of $\alpha_s$ at scale $\sqrt{s}$ can be extracted, as mentioned above. On the contrary, in applying the PMC method, since the PMC scales vary with the value of the event shapes *T* and *C*, we can extract $\alpha_s(Q^2)$ over a wide range of $Q^2$ using the experimental data at a single energy of $\sqrt{s}$. By comparing PMC predictions with measurements at $\sqrt{s} = 91.2$ GeV, we present the extracted running coupling $\alpha_s(Q^2)$ from the thrust and *C*-parameter distributions in Figure 4.

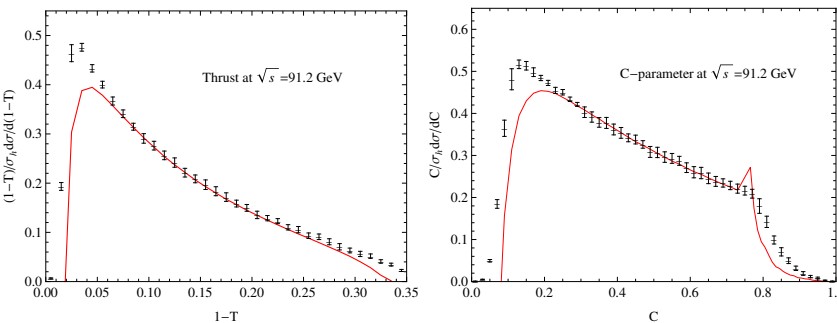

**Figure 3.** The thrust (*T*) and *C*-parameter (*C*) distributions using PMC scale setting for $\sqrt{s} = 91.2$ GeV. The experimental data are taken from the ALEPH Collaboration [39].

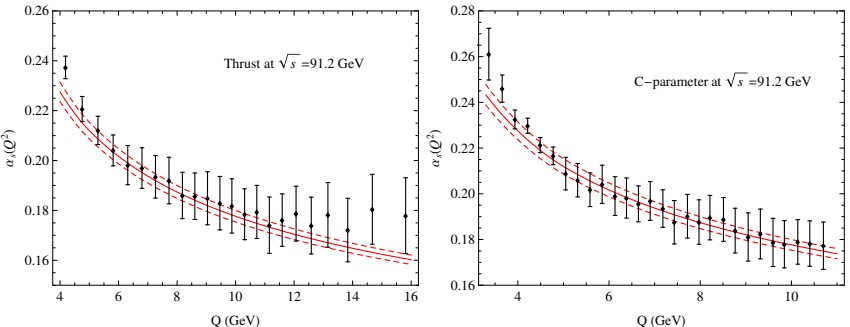

**Figure 4.** The extracted running coupling $\alpha_s(Q^2)$ from the thrust (*T*) and *C*-parameter (*C*) distributions by comparing the PMC predictions with the ALEPH data [39] measured at a single energy of $\sqrt{s} = 91.2$ GeV. As a comparison, the three lines represent the world average [38].

Figure 4 shows that the extracted $\alpha_s(Q^2)$ in the ranges $4 < Q < 16$ GeV from the thrust and $3 < Q < 11$ GeV from the *C*-parameter are in excellent agreement with the world average evaluated using the value $\alpha_s(M_Z^2) = 0.1179$ [38] for the coupling at $Z^0$ mass. Since the PMC method eliminates the renormalization scale uncertainty, the extracted $\alpha_s(Q^2)$ is not plagued by any uncertainty from the choice of the scale $\mu_r$. Thus, PMC scale setting provides a remarkable way to verify the running of $\alpha_s(Q^2)$ from event shape observables in $e^+e^-$ annihilation measured at a single energy $\sqrt{s}$.

The mean value of event shape observables provides an important complement to the differential distributions and to the determination of $\alpha_s$. The mean value of an event shape *y* is defined as

$$\langle y \rangle \;=\; \int_0^{y_0} \frac{y}{\sigma_h} \frac{d\sigma}{dy} dy, \tag{6}$$

where $y_0$ is the kinematically-allowed upper bound of the *y* variable, and the integration is over the full phase space.

In the case of a conventional scale setting, the mean values of *T* and *C* are plagued by the renormalization scale uncertainties and substantially deviate from the measurements even at NNLO [56,57], similar to the case of the differential distributions.

Currently, the most common way to calculate the integral for the mean values is to distinguish the two perturbative and non-perturbative contributions and to calculate them separately. This is known and extensively studied in the literature. Nevertheless, some artificial parameters and a theoretical model have to be introduced in order to match theoretical predictions with experimental data.

After applying the PMC, we obtain

$$\mu_r^{\text{pmc}}|_{\langle 1-T \rangle} = 0.0695\sqrt{s} \tag{7}$$

for the mean value of the thrust and

$$\mu_r^{\text{pmc}}|_{\langle C \rangle} = 0.0656\sqrt{s} \tag{8}$$

for the mean value of the *C*-parameter. The PMC scales satisfy $\mu_r^{\text{pmc}} \ll \sqrt{s}$, reflecting the soft virtuality of the underlying QCD subprocesses. We note that in the analysis of Ref. [39], using a conventional scale setting leads to an anomalously large value of $\alpha_s$, demonstrating again that the correct description for the mean values requires $\mu_r \ll \sqrt{s}$. The PMC scales for the differential distributions of the thrust and *C*-parameter are also very small. PMC scale setting is self-consistent with the differential distributions of the event shapes and their mean values.

After using PMC scale setting, the thrust and *C*-parameter mean values are increased, especially at small $\sqrt{s}$. The scale-independent PMC predictions are in excellent agreement with the experimental data over a wide range of center-of-mass energies $\sqrt{s}$ [51]. Since we can obtain a high degree of consistency between the PMC predictions and the measurements, the QCD coupling $\alpha_s(Q^2)$ can be extracted with high precision. The extracted QCD coupling $\alpha_s(Q^2)$ in the $\overline{\text{MS}}$ scheme from the thrust and *C*-parameter mean values is presented in Figure 5. This figure shows that the extracted $\alpha_s(Q^2)$ values are mutually compatible and are in excellent agreement with the world average.

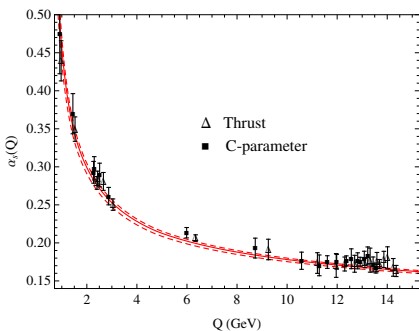

**Figure 5.** The extracted QCD coupling $\alpha_s(Q^2)$ from the thrust and *C*-parameter mean values by comparing PMC predictions with the JADE and OPAL data [41,58]. The error bars are the squared averages of the experimental and theoretical errors. The three lines are the world average [38].

A highly precise determination of the value of $\alpha_s(M_Z^2)$ fitting the PMC predictions to the measurements is achieved. Finally, we obtain [51]

$$\alpha_s(M_Z^2) = 0.1185 \pm 0.0012 \tag{9}$$

from the thrust mean value and

$$\alpha_s(M_Z^2) = 0.1193^{+0.0021}_{-0.0019}, \tag{10}$$

from the *C*-parameter mean value. Since the dominant renormalization scale $\mu_r$ uncertainty is eliminated and the convergence of pQCD series is greatly improved after using the PMC method, the precision of the extracted $\alpha_s$ values is largely improved.

### 3.2. Heavy Quark Pair Production in $e^+e^-$ Annihilation near the Threshold Region

Heavy fermion pair production in $e^+e^-$ annihilation is a fundamental process for the SM and of considerable interest for other phenomena. Heavy quark interaction in the threshold region is of particular interest due to the presence of singular terms from the QCD Coulomb corrections. Physically, the renormalization scale that reflects the subprocess virtuality becomes very soft in this region. It is conventional to set the renormalization scale to the mass of the heavy fermion $\mu_r = m_f$. This conventional procedure obviously violates the physical behavior of the QCD corrections and leads to results affected by systematic

errors, due to inherent scheme and scale uncertainties, and predictions are quite unreliable in this kinematic region.

The quark pair production cross-section for $e^+ e^- \to \gamma^* \to Q\bar{Q}$ at the two-loop level can be written as

$$\sigma = \sigma^{(0)} \left[ 1 + \delta^{(1)} a_s(\mu_r) + \delta^{(2)}(\mu_r) a_s^2(\mu_r) + \mathcal{O}(a_s^3) \right], \tag{11}$$

where $a_s(\mu_r) = \alpha_s(\mu_r)/\pi$. The LO cross section is

$$\sigma^{(0)} = \frac{4}{3} \frac{\pi \alpha_e^2}{s} N_c e_Q^2 \frac{v(3 - v^2)}{2}, \tag{12}$$

where $\alpha_e$ is the fine structure constant, $N_c$ is the number of colors, and $e_Q$ is the $Q$ quark electric charge. The quark velocity $v$ is $v = \sqrt{1 - 4 m_Q^2/s}$, where $s$ is the center-of-mass energy squared and $m_Q$ is the mass of the quark $Q$.

The one-loop correction coefficient $\delta^{(1)}$ is $\delta^{(1)} = C_F(\pi^2/2v - 4)$. The two-loop correction coefficient $\delta^{(2)}$ can be conveniently split into terms proportional to different $SU(3)$ color factors,

$$\begin{aligned} \delta^{(2)} &= C_F^2 \delta_A^{(2)} + C_F C_A \delta_{NA}^{(2)} \\ &+ C_F T_R n_f \delta_L^{(2)} + C_F T_R \delta_H^{(2)}. \end{aligned} \tag{13}$$

The terms $\delta_A^{(2)}$, $\delta_L^{(2)}$, and $\delta_H^{(2)}$ are the same in Abelian and non-Abelian theories; the term $\delta_{NA}^{(2)}$ only arises in the non-Abelian theory. This process offers the opportunity to rigorously rigorously the scale-setting method in the non-Abelian and Abelian theories.

The cross-section given in Equation (11) can be written indicating explicitly the $n_f$-dependent and $n_f$-independent parts, i.e.,

$$\begin{aligned} \sigma &= \sigma^{(0)} \left[ 1 + \delta_h^{(1)} a_s(\mu_r) + \left( \delta_{h,in}^{(2)}(\mu_r) + \delta_{h,n_f}^{(2)}(\mu_r) n_f \right) a_s^2(\mu_r) \right. \\ &\left. + \left( \frac{\pi}{v} \right) \delta_v^{(1)} a_s(\mu_r) + \left( \frac{\pi}{v} \right) \left( \delta_{v,in}^{(2)}(\mu_r) + \delta_{v,n_f}^{(2)}(\mu_r) n_f \right) a_s^2(\mu_r) + \left( \frac{\pi}{v} \right)^2 \delta_{v^2}^{(2)} a_s^2(\mu_r) + \mathcal{O}(a_s^3) \right]. \end{aligned} \tag{14}$$

Coefficients $\delta_v^{(1)}$, $\delta_v^{(2)}$, and $\delta_{v^2}^{(2)}$ are for the Coulomb corrections, while coefficients $\delta_h^{(1)}$ and $\delta_h^{(2)}$ are for the non-Coulomb corrections. These coefficients have been determined in Refs. [59–61] for the $\overline{\text{MS}}$ scheme. Due to their proportional form to powers of $(\pi/v)$, Coulomb corrections are enhanced in the threshold region. This implies that the renormalization scale can be relatively soft in this region. Therefore, the PMC scales must be determined separately for the non-Coulomb and Coulomb corrections [8,62]. When the quark velocity $v \to 0$, the Coulomb correction dominates the contribution for the production cross-section.

Absorbing the non-conformal term $\beta_0 = 11/3\, C_A - 4/3\, T_R n_f$ into the running coupling constant, as implemented in the PMC procedure, we obtain

$$\begin{aligned} \sigma &= \sigma^{(0)} \left[ 1 + \delta_h^{(1)} a_s(Q_h) + \delta_{h,\text{sc}}^{(2)}(\mu_r) a_s^2(Q_h) \right. \\ &+ \left( \frac{\pi}{v} \right) \delta_v^{(1)} a_s(Q_v) + \left( \frac{\pi}{v} \right) \delta_{v,\text{sc}}^{(2)}(\mu_r) a_s^2(Q_v) \\ &\left. + \left( \frac{\pi}{v} \right)^2 \delta_{v^2}^{(2)} a_s^2(Q_v) + \mathcal{O}(a_s^3) \right]. \end{aligned} \tag{15}$$

The PMC scales $Q_i$ are

$$Q_i = \mu_r \exp\left[\frac{3\,\delta_{i,n_f}^{(2)}(\mu_r)}{2\,T_R\,\delta_i^{(1)}}\right],$$
(16)

and the coefficients $\delta_{i,\text{sc}}^{(2)}(\mu_r)$ are

$$\delta_{i,\text{sc}}^{(2)}(\mu_r) = \frac{11\,C_A\,\delta_{i,n_f}^{(2)}(\mu_r)}{4\,T_R} + \delta_{i,in}^{(2)}(\mu_r),$$
(17)

where $i = h$ and $v$ stand for the non-Coulomb and Coulomb corrections, respectively. At the present two-loop level, the conformal coefficients and the PMC scales are independent of the renormalization scale $\mu_r$. Thus, the resulting cross-section in Equation (16) is void of renormalization scale uncertainties.

The V-scheme defined by the interaction potential between two heavy quarks [63–71], $V(Q^2) = -4\,\pi^2\,C_F\,a_s^V(Q)/Q^2$, provides a physically-based alternative to the usual $\overline{\text{MS}}$ scheme. As in the case of QED, when the scale of the coupling $a_s^V$ is identified with the exchanged momentum, all vacuum polarization corrections are resummed into $a_s^V$. By using the relation between $a_s$ and $a_s^V$ at the one-loop level, i.e.,

$$a_s^V(Q) = a_s(Q) + \left(\frac{31}{36}C_A - \frac{5}{9}T_R\,n_f\right)a_s^2(Q),$$
(18)

we can perform a change of scheme, from the $\overline{\text{MS}}$ scheme to the V-scheme, for the quark pair production cross-section. The corresponding perturbative coefficients in Equation (14) for the V-scheme are given in Ref. [72]. The predictions using the PMC eliminate the dependence from the renormalization scheme; this is explicitly displayed in the form of "commensurate scale relations" (CSR) [73,74].

The PMC scales in the $\overline{\text{MS}}$ scheme are $Q_h = e^{(-11/24)}\,m_Q$ for the non-Coulomb correction and $Q_v = 2\,e^{(-5/6)}\,v\,m_Q$ for the Coulomb correction. In the V-scheme, the scales are $Q_h = e^{(3/8)}\,m_Q$ for the non-Coulomb correction and $Q_v = 2\,v\,m_Q$ for the Coulomb correction. The scale $Q_h$ stems from the hard virtual gluon corrections, and the scale $Q_v$ originates from the final state Coulomb re-scattering. As expected, the scale $Q_h$ is of the order $m_Q$, whereas the scale $Q_v$ is of the order $v\,m_Q$. The scale $Q_v$ depends on the quark velocity $v$ and becomes soft for $v \to 0$, yielding the correct physical behavior. The PMC scales in the usual $\overline{\text{MS}}$ scheme are different from the PMC scales in the physically-based V-scheme. This difference is caused by the convention used in defining the $\overline{\text{MS}}$ scheme.

For the Coulomb correction, the behavior of the Coulomb term of the form $(\pi/v)\,\delta_v^{(2)}$ is dramatically changed after using the PMC. More explicitly, by taking $m_Q = 4.89$ GeV for the $b$ quark pair production as an example, we present the behavior of the Coulomb terms of the form $(\pi/v)\,\delta_v^{(2)}$ in the V-scheme using the conventional and the PMC scale setting in Figure 6. Using the conventional scale setting in the region where the quark velocity $v \to 0$, the Coulomb term becomes $(\pi/v)\delta_v^{(2)} \to +\infty$. On the contrary, applying PMC scale setting, the Coulomb term becomes $(\pi/v)\delta_v^{(2)} \to -\infty$. This dramatically different behavior of the $(\pi/v)\delta_v^{(2)}$ between conventional and PMC scale settings near the threshold region should also be investigated in QED.

In analogy to the quark pair production, the lepton pair production cross-section for the QED process $e^+e^- \to \gamma^* \to l\bar{l}$ has an expansion in the QED fine structure constant $\alpha_e$. The cross-section can also be divided into the non-Coulomb and Coulomb parts, as in Equation (14). The perturbative coefficients for the lepton pair production cross-section are given in Refs. [59,75,76].

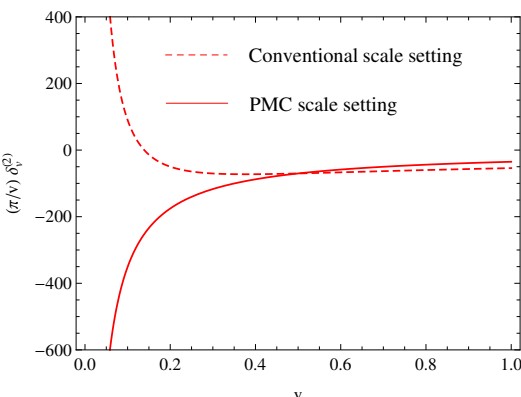

**Figure 6.** The behavior of the Coulomb terms in the V-scheme for the *b* quark pair production, where $\delta_v^{(2)} = (\delta_{v,in}^{(2)}|_V + \delta_{v,n_f}^{(2)}|_V n_f)$ for conventional scale setting, and for PMC scale setting, $\delta_v^{(2)} = \delta_{v,sc}^{(2)}|_V$.

The one-loop correction coefficients $\delta_h^{(1)}$ and $\delta_v^{(1)}$ and the two-loop correction coefficients $\delta_{h,n_f}^{(2)}$, $\delta_{v,n_f}^{(2)}$, and $\delta_{v^2}^{(2)}$ have the same form in QCD and QED, with only some replacements for the color factors, i.e., $C_A = 3$, $C_F = 4/3$ and $T_R = 1/2$ for QCD and $C_A = 0$, $C_F = 1$, and $T_R = 1$ for QED, respectively.

By using the PMC, the vacuum polarization corrections can be absorbed into the QED running coupling whose one-loop approximation is given by:

$$\alpha_e(Q) = \alpha_e \left[ 1 + \left( \frac{\alpha_e}{\pi} \right) \sum_{i=1}^{n_f} \frac{1}{3} \left( \ln \left( \frac{Q^2}{m_i^2} \right) - \frac{5}{3} \right) \right], \tag{19}$$

where $m_i$ is the mass of the light virtual lepton. The resulting PMC scales can be written as

$$Q_i = m_l \exp \left[ \frac{5}{6} + \frac{3}{2} \frac{\delta_{i,n_f}^{(2)}}{\delta_i^{(1)}} \right]. \tag{20}$$

For the lepton pair production, we obtain the PMC scales $Q_h = e^{(3/8)} m_l$ for the non-Coulomb correction and $Q_v = 2 v m_l$ for the Coulomb correction.

Given that the scale $Q_h$ stems from the hard virtual photon corrections, while $Q_v$ originates from the Coulomb rescattering, it follows that $Q_h$ is of order $m_l$ and $Q_v$ is of order $v m_l$. The scales show the same physical behavior from QCD to QED after using the PMC. The PMC scales in QCD with the V-scheme coincide with the PMC scales in QED. This scale self-consistency shows that the PMC procedure in QCD agrees with the standard Gell-Mann–Low method [4] in QED for the quark pair production.

For the Coulomb correction, by taking $m_\tau = 1.777$ GeV for the $\tau$ lepton as an example, the behavior of the Coulomb terms of the form $(\pi/v) \delta_v^{(2)}$ using conventional and PMC scale settings is shown in Figure 7. It is noted that, different from the QCD case, when the quark velocity $v \to 0$, the Coulomb terms are $(\pi/v)\delta_v^{(2)} \to -\infty$ for both the conventional and the PMC scale settings in lepton pair production. Thus, the behavior of the Coulomb terms is the same when using PMC scale setting for both QCD and QED.

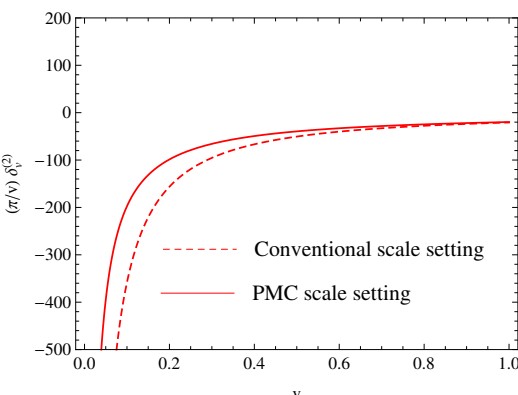

**Figure 7.** The behavior of the Coulomb terms for the $\tau$ lepton pair production, where $\delta_v^{(2)} = (\delta_{v,in}^{(2)} + \delta_{v,n_f}^{(2)} n_f)$ for conventional scale setting, and for PMC scale setting, $\delta_v^{(2)} = \delta_{v,in}^{(2)}$.

*3.3. TReanalysis of the Top-Quark Decay at Next-To-Next-To-Leading Order*

Top-quark properties, such as its mass, its production cross-section, its decay width, and its couplings to elementary particles, are very important for understanding the mechanism of electro-weak symmetry breaking and for searching new physics beyond the SM. At present, the top-quark decay width has been calculated up to NNLO QCD corrections [77–82]. Experimentally, various collaborations at the Tevatron and LHC have measured the total width of the top-quark decay, and the world average reported by the Particle Data Group is $\Gamma_t = 1.42^{+0.19}_{-0.15}$ GeV [38].

The top-quark decay process is dominated by $t \to bW$, and its total decay width up to NNLO QCD corrections is given by:

$$\Gamma_t = \Gamma_t^{\text{LO}}\left[1 + c_1(\mu_r)\, a_s(\mu_r) + c_2(\mu_r)\, a_s^2(\mu_r) + \mathcal{O}(\alpha_s^3)\right], \tag{21}$$

where the LO decay width

$$\Gamma_t^{\text{LO}} = \frac{G_F\, |V_{tb}|^2\, m_t^3}{8\,\pi\,\sqrt{2}}\left(1 - 3\,w^2 + 2\,w^3\right). \tag{22}$$

Here, $w = m_W^2/m_t^2$, with $m_W = 80.385$ GeV being the $W$-boson mass, $m_t = 172.5$ GeV [38] is the top-quark mass, $G_F = 1.16638 \times 10^{-5}$ GeV$^{-2}$ [81] is the Fermi constant, and $|V_{tb}| = 1$ is the Cabibbo–Kobayashi–Maskawa (CKM) matrix element. The NLO and NNLO coefficients $c_1$ and $c_2$ can be found in the literature, and a detailed PMC analysis for the top-quark decay process can be found in Ref. [24]. For self-consistency, we will adopt the two-loop $\overline{\text{MS}}$ QCD coupling with $\alpha_s(M_Z) = 0.1179$ [38] for numerical analysis.

We present the total decay width of the top-quark decay using the conventional and PMC scale settings in Table 1, where the NLO and NNLO contributions $\delta\Gamma_t^{\text{NLO}}$ and $\delta\Gamma_t^{\text{NNLO}}$ are also shown. Up to the NNLO level, the net scale uncertainty is $\sim [-0.5\%, +0.4\%]$ by varying the scale $\mu_r$ within the range $[m_t/2, 2m_t]$. Such a small net scale uncertainty is due to the cancellation of the scale uncertainties between $\delta\Gamma_t^{\text{NLO}}$ and $\delta\Gamma_t^{\text{NNLO}}$. However, the scale uncertainty is still rather large for each perturbative term, i.e., the scale uncertainties of $\delta\Gamma_t^{\text{NLO}}$ and $\delta\Gamma_t^{\text{NNLO}}$ are $\sim [-10.5\%, +7.9\%]$ and $\sim [+23.5\%, -16.7\%]$, respectively.

If we set $\mu_r = m_t$, the relative importance of the NLO and NNLO QCD corrections become $\delta\Gamma_t^{\text{NLO}}/\Gamma_t^{\text{LO}} \sim -8.6\%$ and $\delta\Gamma_t^{\text{NNLO}}/\Gamma_t^{\text{LO}} \sim -2.1\%$, respectively. The relative importance of each order of accuracy up to NNLO becomes: $\delta\Gamma_t^{\text{NLO}}/\Gamma_t^{\text{LO}} \sim -9.4\%$ and $\delta\Gamma_t^{\text{NNLO}}/\Gamma_t^{\text{LO}} \sim -1.6\%$ for $\mu_r = m_t/2$; and $\delta\Gamma_t^{\text{NLO}}/\Gamma_t^{\text{LO}} \sim -7.8\%$ and $\delta\Gamma_t^{\text{NNLO}}/\Gamma_t^{\text{LO}} \sim -2.4\%$ for $\mu_r = 2m_t$. Thus, by using the conventional scale-setting method, one cannot judge a posteriori the intrinsic convergence of the pQCD series; a poorer convergent behavior may be caused by improper choice of renormalization scale. This explains why the renormalization scale uncertainty is one of the systematic errors for pQCD predictions.

**Table 1.** Total decay width $\Gamma_t$ up to NNLO QCD corrections (in unit GeV) using the conventional (Conv.) and PMC scale settings, respectively.

| | Scale $\mu_r$ | $\Gamma_t^{LO}$ | $\delta\Gamma_t^{NLO}$ | $\delta\Gamma_t^{NNLO}$ | $\Gamma_t^{NNLO}$ |
|---|---|---|---|---|---|
| | $\mu_r = m_t/2$ | 1.4806 | $-0.1394$ | $-0.0234$ | 1.3179 |
| Conv. | $\mu_r = m_t$ | 1.4806 | $-0.1261$ | $-0.0306$ | 1.3239 |
| | $\mu_r = 2m_t$ | 1.4806 | $-0.1161$ | $-0.0357$ | 1.3288 |
| PMC | | 1.4806 | $-0.1892$ | 0.0207 | 1.3122 |

On the other hand, after applying the PMC scale setting, Table 1 shows that the PMC predictions are scale invariant, e.g., $\delta\Gamma_t^{NLO} = -0.1892$ GeV and $\delta\Gamma_t^{NNLO} = 0.0207$ GeV for any choice of $\mu_r$. This leads to a scale invariant relative importance of the NLO and NNLO QCD corrections, e.g., $\delta\Gamma_t^{NLO}/\Gamma_t^{LO} \sim -12.8\%$ and $\delta\Gamma_t^{NNLO}/\Gamma_t^{LO} \sim 1.4\%$ for any choice of scale. Thus, with respect to the conventional pQCD series for the top-quark decay, the convergence of the PMC series is greatly improved.

The determined PMC scale for the top-quark decay is

$$Q = 15.5 \text{ GeV}. \tag{23}$$

The PMC scale is independent of any choice of $\mu_r$ and is one order of magnitude smaller than $m_t$. This reflects the small virtuality of the QCD dynamics for the top-quark decay process. Numerically, we observe that the top-quark decay width at NNLO first decreases and then increases with the increase in the scale $\mu_r$ using a conventional scale setting, and the minimum total decay width is achieved at $\mu_r \sim 23$ GeV. If we change the conventional choice $\mu_r = m_t$ to a smaller-scale $\mu_r \ll m_t$, the pQCD convergence of the top-quark decay width would be greatly improved, even though the resulting conventional prediction is close to the PMC prediction. Thus, the effective momentum flow for the top-quark decay should be $\ll m_t$, far smaller than the conventional choice of $\mu_r = m_t$.

After applying the PMC in order to achieve reliable predictions, there are still other error sources that have to be taken into account, such as the effect of finite bottom-quark mass and the finite $W$ boson width, as well as the electroweak corrections. In Table 2, we present the top-quark decay width $\Gamma_t^{NNLO}|_{PMC}$ using the PMC together with the corrections from the finite bottom-quark mass $\delta_f^b$, the finite $W$ boson width $\delta_f^W$, and the NLO electroweak correction $\delta_{EW}^{NLO}$ for $m_t = 172.5$ and 173.5 GeV. These corrections are taken from Ref. [81]. Since the corrections from the finite bottom-quark mass and the finite $W$ boson width provide negative values, while the NLO electroweak correction provides a positive value, their contributions to the top-quark decay width cancel out greatly.

**Table 2.** PMC top-quark decay widths $\Gamma_t^{NNLO}|_{PMC}$ (in unit GeV). Uncertainties caused by the bottom-quark mass $\delta_f^b$, the finite $W$-boson width $\delta_f^W$, and the NLO electroweak corrections $\delta_{EW}^{NLO}$ are also presented, whose magnitudes are taken from Ref. [81].

| $m_t$ | $\Gamma_t^{NNLO}|_{PMC}$ | $\delta_f^b$ | $\delta_f^W$ | $\delta_{EW}^{NLO}$ | $\Gamma_t^{tot}$ |
|---|---|---|---|---|---|
| 172.5 | 1.3122 | $-0.0038$ | $-0.0221$ | 0.0249 | 1.3112 |
| 173.5 | 1.3392 | $-0.0039$ | $-0.0225$ | 0.0255 | 1.3383 |

After applying the PMC, we then obtain more reliable predictions for the top-quark total decay width [24]. If we set $m_t = 172.5$ GeV, we have

$$\Gamma_t^{tot} = 1.3112 \pm 0.0016 \pm 0.0023 \text{ GeV}, \tag{24}$$

whhereas if we set $m_t = 173.5$ GeV, we have

$$\Gamma_t^{tot} = 1.3383^{+0.0016}_{-0.0017} \pm 0.0023 \text{ GeV}. \tag{25}$$

The first error is due to the error of $\alpha_s$ at the critical scale $\mu_r = M_Z$, e.g., $\Delta\alpha_s(M_Z) = \pm0.0009$ [38], and the second error is given by the evaluation of the UHO terms. The top-quark total decay width depends heavily on the magnitude of the top-quark mass. More explicitly, we present the top-quark total decay width $\Gamma_t^{\text{tot}}$ versus $m_t$ in Figure 8. The CMS measurement together with its error [83] are also presented in Figure 8. No discrepancy is observed in comparing the experimental value and the theoretical predictions obtained by using the PMC and the conventional scale setting, respectively (Figure 8).

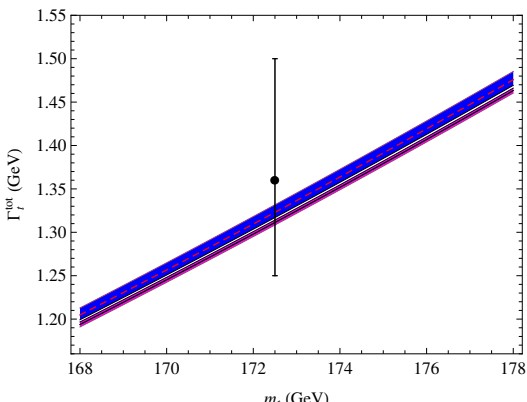

**Figure 8.** The top-quark total decay width $\Gamma_t^{\text{tot}}$ versus the top-quark mass $m_t$. The solid line is the PMC prediction, and the dashed line stands for the conventional prediction. As a comparison, the CMS measured value [83] is also presented.

### 3.4. An Estimate of the Contributions from Uncalculated Higher-Order Terms

At present, remarkable progress has been achieved in loop calculations in perturbation theory. However, most of theoretical calculations have only been finished at relatively lower orders due to the complexity of Feynman diagram calculations. It is thus important to have a way to evaluate contributions from the UHO terms, in order to improve the predictive power of perturbative theory.

In this subsection, we briefly review two representative approaches to evaluate the UHO contributions for the fixed-order pQCD series by using the known partial sum of the conventional series and PMC conformal series, respectively. The first approach is the Padé approximation approach (PAA) [84–86], which attempts to directly predict the unknown higher-order coefficients by using a fractional generating function whose parameters can be directly fixed by matching to the known finite order. The second approach is the Bayesian-based approach (BA) [87–90], which attempts to quantify the unknown higher-order terms in terms of a probability distribution by applying Bayes' theorem.

#### 3.4.1. Applying PAA to Evaluate the UHO Contributions

The Padé approximation provides a feasible approach that predicts the unknown $(n+1)_{\text{th}}$-order coefficient from the known $n_{\text{th}}$-order perturbative series. Following the basic PAA procedure, an $[N/M]$-type fractional generating function $\rho_n^{[N/M]}$ for $\rho_n = \sum_{i=0}^{n(\geq1)} c_i \alpha_s^i$ is constructed as [84–86]

$$\rho_n^{[N/M]} = \frac{d_0 + d_1\alpha_s + \cdots + d_N\alpha_s^N}{1 + e_1\alpha_s + \cdots + e_M\alpha_s^M}$$

$$= \sum_{i=0}^{n} c_i\alpha_s^i + c_{n+1}\alpha_s^{n+1} + \cdots , \tag{26}$$

where $M \geq 1$ and $N + M = n$. The parameters $d_i$ $(0 \leq i \leq N)$ and $e_j$ $(1 \leq j \leq M)$ are firstly determined by the known coefficients $c_i$ $(0 \leq i \leq n)$ and then provide a reasonable prediction for the next uncalculated coefficient $c_{n+1}$. For $n = 4$, it has been found that the diagonal $[2/2]$-type generating function is preferable for predicting unknown

coefficients from the conventional pQCD series [91,92], while the non-diagonal [0/4]-type generating function is preferable for predicting unknown coefficients from the PMC conformal series [93], which also expands the geometric series to be self-consistent with the GM-L prediction [4].

### 3.4.2. Applying BA to Evaluate the UHO Contributions

The BA quantifies the UHO coefficients in terms of probability distributions, in which Bayes' theorem is applied to iteratively update the probability as new coefficients become available. Here we present the main results for BA; for a detailed introduction and all BA formulas, see, e.g., [94] and references therein.

Following the BA procedure, the conditional probability density function (p.d.f.) for a generic (uncalculated) coefficient $c_n$ ($n > k$) of any possible perturbative series $\rho_k = \sum_{i=1}^{k} c_i \alpha_s^i$ with given coefficients $\{c_1, c_2, \ldots, c_k\}$ is given by

$$f(c_n | c_1, c_2, \ldots, c_k) = \begin{cases} \frac{k}{2(k+1)\bar{c}_{(k)}}, & |c_n| \leq \bar{c}_{(k)} \\ \frac{k\bar{c}_{(k)}^k}{2(k+1)|c_n|^{k+1}}, & |c_n| > \bar{c}_{(k)} \end{cases}, \tag{27}$$

where $\bar{c}_{(k)} = \max\{|c_1|, |c_2|, \cdots, |c_k|\}$. Equation (27) provides a symmetric probability distribution for negative and positive $c_n$, predicts a uniform probability density in the interval $[-\bar{c}_{(k)}, \bar{c}_{(k)}]$, and decreases monotonically from $\bar{c}_{(k)}$ to infinity. The knowledge of probability density $f_c(c_n | c_1, c_2, \ldots, c_k)$ allows one to calculate the degree-of-belief (DoB) that the value of $c_n$ belongs to some credible interval (CI). The symmetric smallest CI of fixed $p\%$ DoB for $c_n$ is denoted by $[-c_n^{(p)}, c_n^{(p)}]$. Here the boundary $c_n^{(p)}$ is defined implicitly by $p\% = \int_{-c_n^{(p)}}^{c_n^{(p)}} f_c(c_n | c_l, \ldots, c_k) \, dc_n$ and can be obtained by further using the analytical expression in Equation (27),

$$c_n^{(p)} = \begin{cases} \bar{c}_{(k)} \frac{k+1}{k} p\%, & p\% \leq \frac{k}{k+1} \\ \bar{c}_{(k)} [(k+1)(1-p\%)]^{-\frac{1}{k}}, & p\% > \frac{k}{k+1} \end{cases}. \tag{28}$$

We adopt the interval $[-c_n^{(p)} \alpha_s^n, c_n^{(p)} \alpha_s^n]$ with $p\% = 95.5\%$[1] as the final estimation for any UHO term $\delta_n = c_n \alpha_s^n$.

As an example, we consider the total hadronic $e^+ e^-$ annihilation ratio $R_{e^+ e^-}(Q) = N_c \sum_q e_q^2 [1 + R(Q)]$, where $R(Q)$ represents the QCD correction. The probability density distributions for $R(Q = 31.6 \text{ GeV})$ with different states of knowledge predicted by PMCs and BA are presented in Figure 9, where the four curves correspond to different degrees of knowledge: given LO (dotted), given NLO (dot-dashed), given N²LO (solid), and given N³LO (dashed). The figure illustrates the characteristics of the posterior probability distribution: a symmetric plateau with two suppressed tails. The posterior probability distribution depends on the prior probability distribution. With more and more loop terms available, the posterior probability is updated and becomes less and less dependent on the prior probability; i.e., the probability density becomes increasingly concentrated as more and more loop terms are added.

As a final remark, the PAA and BA can only be applied after one has specified the choice for the renormalization scale due to the fact that the coefficients of the conventional pQCD series are scale-dependent. Thus, extra uncertainties are introduced when applying the PAA and BA to a conventional pQCD series. However, the resulting PMC series is scale-independent, and it thus provides a more reliable basis for estimating the UHO contributions. Thus, the total theoretical uncertainty of a pQCD approximant can be treated as the squared average of the scale error due to the conventional scale dependence (or the first kind of residual scale dependence) and the predicted magnitude of the UHO terms for the pQCD approximant.

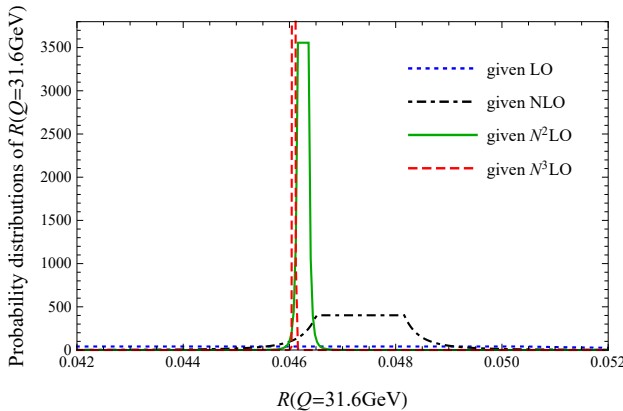

**Figure 9.** The probability density distributions of $R(Q = 31.6 \text{ GeV})$ with different states of knowledge predicted by PMCs and BA. The blue dotted, the black dash-dotted, the green solid, and the red dashed curves represent the results with given LO, NLO, N$^2$LO, and N$^3$LO series, respectively.

## 4. Summary

The PMC method provides a systematic way to eliminate the renormalization scheme-and-scale ambiguities. The PMC method has a rigorous theoretical foundation, satisfying the RGI and all of the self-consistency conditions derived from the renormalization group. The PMC scales are obtained by shifting the argument of $\alpha_s$ to eliminate all the non-conformal $\beta$-terms; the PMC scales thus reflect the virtuality of the propagating gluons for the QCD processes. The divergent renormalon contributions are eliminated since they are summed into the running coupling $\alpha_s$, and the resulting pQCD convergence is in general greatly improved. The PMC scale-setting method provides the underlying principle for the well-known BLM method, extending the BLM scale-setting procedure unambiguously to all orders. The PMC reduces to the GM-L method in the $N_C \to 0$ Abelian limit [5].

We have provided a new analysis of event shape observables in $e^+e^-$ annihilation by using the PMC method. The PMC scales are not given by a single value but depend dynamically on the virtuality of the underlying quark and gluon subprocess and thus the specific kinematics of each event. The renormalization scale-independent PMC predictions for event shape distributions agree with precise experimental data. Remarkably, the PMC method provides a novel method for the precise determination of the running of QCD coupling $\alpha_s(Q^2)$ over a wide range of $Q^2$ from event shapes measured at a single energy of $\sqrt{s}$. The PMC also provides an unambiguous method for determining the scales in multiple-scale processes. It is remarkable that two distinctly different PMC scales are determined for the heavy fermion pair production near the threshold region. One PMC scale entering the hard virtual corrections is of the order of the fermion mass $m_f$, while the other PMC scale entering the Coulomb re-scattering amplitude is of the order $v\, m_f$. Perfect agreement between the Abelian unambiguous Gell-Mann-Low and the PMC scale-setting method in the limit of zero number of colors is observed in this process. We also calculated the top-quark decay process, obtaining the PMC scale $Q = 15.5$ GeV. The convergence of the pQCD series is largely improved for the top-quark decay. We finally obtained the top-quark total decay width $\Gamma_t^{\text{tot}} = 1.3112^{+0.0190}_{-0.0189}$ GeV. Since the PMC conformal series is scale-independent, it provides a reliable basis for obtaining constraints on the predictions for the UHO contributions. These applications demonstrate the generality and applicability of the PMC. The PMC thus improves the precision tests of the SM and and increases the sensitivity of experiments to new physics beyond the SM.

**Funding:** This work was supported in part by the Natural Science Foundation of China under Grants No.12265011, No.12175025, and No.12147102; by the Project of Guizhou Provincial Department under Grants No.KY[2021]003, No.GZMUZK[2022]PT01, and No.ZK[2023]141; and by the Department of Energy (DOE), Contract DECAC02C76SF00515. SLAC-PUB-17723.

**Conflicts of Interest:** The authors declare no conflict of interest.

## Note

1    One may also use a 68.3% credible interval (CI) to compare with experimental data in the same confidence level, or use 99.7% CI for a more conservative estimation.

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
