# Peer review of "Elimination of QCD Renormalization Scale and Scheme Ambiguities"

_universe, doi:10.3390/universe9040193_

Round 1
Reviewer 1 Report
In this article a mini review of the PMC method, Principle of Maximum Conformality, a systematic way to eliminate the renormalization scheme-and-scale dependence, and its recent applications, are presented. The paper shows that the PMC method can be used to a large extend of observables. The evaluation of uncertainties related to higher order terms is also summarized. The top quark decay process is calculated and the improvement of the convergence of the pQCD series is shown. The present article has made a good review of the status of the PMC method, showing its range of applicability, what can be useful for future applications of precision tests of the SM. For these reasons I can recommend this article in the present form.
Author Response
We thank the referee for his/her careful reading of our paper and for his/her positive comments on our manuscript, ``Elimination of QCD Renormalization Scale and Scheme Ambiguities". In the revised version, we have made some corrections to reduce the similarity rate and the self-citation rate which have lead to an improved and more readable manuscript.
Reviewer 2 Report
This paper is devoted to one of the central aspects related to pQCD, namely the renormalization scale setting. The main point is the elimination of the scale and scheme ambiguities, present in the conventional approach, by means of a formalism based on the Principle of Maximum Conformality (PMC).
After reviewing the PMC scale-setting method (both multi-scale and single-scale approaches), with useful references to successfully consequences and results, the authors present some recent applications to important physical processes: event shape observables in e-e+ annihilation, heavy quark pair production, top-quark decay and an interesting discussion on the uncalculated high-order contributions in pQCD.
The paper is highly relevant and timely, deserving publication in the special issue `The Quantum Chromodynamics: 50th Anniversary of the Discovery`.
Misprint in line 758: and and.
Author Response

(The authors gave the same response as above.)

Reviewer 3 Report
This is a nice review of PMC approach. I see no need for revision.
Author Response

(The authors gave the same response as above.)

Reviewer 4 Report
The paper is well-organized, nicely written, and worth to publish in the present form.
Author Response

(The authors gave the same response as above.)
